# A Shared Pathogenic Mechanism for Valproic Acid and *SHROOM3* Knockout in a Brain Organoid Model of Neural Tube Defects

**DOI:** 10.3390/cells12131697

**Published:** 2023-06-23

**Authors:** Taylor N. Takla, Jinghui Luo, Roksolana Sudyk, Joy Huang, John Clayton Walker, Neeta L. Vora, Jonathan Z. Sexton, Jack M. Parent, Andrew M. Tidball

**Affiliations:** 1Department of Neurology, Medical School, University of Michigan, Ann Arbor, MI 48109, USArsudyk@umich.edu (R.S.);; 2Department of Obstetrics and Gynecology, Division of Maternal-Fetal Medicine, School of Medicine, University of North Carolina, Chapel Hill, NC 27599, USA; 3Department of Internal Medicine, Medical School, University of Michigan, Ann Arbor, MI 48109, USA; 4Department of Medicinal Chemistry, College of Pharmacy, University of Michigan, Ann Arbor, MI 48109, USA; 5Center for Drug Repurposing, University of Michigan, Ann Arbor, MI 48109, USA; 6Michigan Neuroscience Institute, Medical School, University of Michigan, Ann Arbor, MI 48109, USA; 7VA Ann Arbor Healthcare System, Ann Arbor, MI 48105, USA

**Keywords:** neuroteratogen, anencephaly, induced pluripotent stem cells

## Abstract

Neural tube defects (NTDs), including anencephaly and spina bifida, are common major malformations of fetal development resulting from incomplete closure of the neural tube. These conditions lead to either universal death (anencephaly) or severe lifelong complications (spina bifida). Despite hundreds of genetic mouse models of neural tube defect phenotypes, the genetics of human NTDs are poorly understood. Furthermore, pharmaceuticals, such as antiseizure medications, have been found clinically to increase the risk of NTDs when administered during pregnancy. Therefore, a model that recapitulates human neurodevelopment would be of immense benefit to understand the genetics underlying NTDs and identify teratogenic mechanisms. Using our self-organizing single rosette cortical organoid (SOSR-COs) system, we have developed a high-throughput image analysis pipeline for evaluating the SOSR-CO structure for NTD-like phenotypes. Similar to small molecule inhibition of apical constriction, the antiseizure medication valproic acid (VPA), a known cause of NTDs, increases the apical lumen size and apical cell surface area in a dose-responsive manner. GSK3β and HDAC inhibitors caused similar lumen expansion; however, RNA sequencing suggests VPA does not inhibit GSK3β at these concentrations. The knockout of *SHROOM3*, a well-known NTD-related gene, also caused expansion of the lumen, as well as reduced f-actin polarization. The increased lumen sizes were caused by reduced cell apical constriction, suggesting that impingement of this process is a shared mechanism for VPA treatment and *SHROOM3*-KO, two well-known causes of NTDs. Our system allows the rapid identification of NTD-like phenotypes for both compounds and genetic variants and should prove useful for understanding specific NTD mechanisms and predicting drug teratogenicity.

## 1. Introduction

Open neural tube defects (NTDs) are common congenital malformations (~1/1000 live births worldwide) and include anencephaly and spina bifida [1]. These major malformations of neurodevelopment lead to spontaneous abortion or neonatal death in the case of anencephaly, and paraplegia or severe physical or cognitive disabilities for spina bifida. These malformations arise from genetic defects or environmental exposures, including toxins, pharmaceuticals, and nutrient deprivation. The need for robust teratogen screening of novel therapeutics was highlighted by the thalidomide tragedy of the 1950s. This compound was marketed as a treatment for morning sickness during pregnancy. Although rodent testing did not indicate teratogenic risk of thalidomide use, this drug unexpectedly led to a dramatic number of congenital malformations, including NTDs, in human fetuses [2]. Despite this tragedy, rodent models remain the standard method for teratogenic screening, even though the problem of species-specific differences persists. Overall, rodent models of drug toxicity have shown a low rate of concordance with humans, leading to the termination of many clinical trials due to human-specific toxicities [3]; neurological toxicities are the most common cause of clinical trial termination (22%). In addition to the species-specific problems with current teratogenicity screening, these models are costly, labor intensive, and result in a moderate rate of false positives and false negatives [4]. For these reasons, President Biden signed the “FDA Modernization Act 2.0” on 29 December 2022, which removes the mandate for animal testing for new drugs and specifically states that alternative testing methods should include “cell-based assays, organ chips, and microphysiological systems” [5].

The first pharmaceutical known to produce congenital malformations in humans was the chemotherapy drug aminopterin [6], which blocks the folic acid pathway. Folic acid deficiency is the most prominent identified risk factor of congenital malformations [7]. Other commonly used pharmaceuticals associated with NTDs are valproic acid (VPA), primarily an antiseizure medication; lithium, a mood stabilizer used for bipolar disorder and major depressive disorder; and possibly HIV antiretroviral therapies [8,9,10,11]. Women taking VPA have a 10-fold increased risk of NTD pregnancies [8,12]. Since its therapeutic discovery, VPA has been found to act on several molecular targets, including as an inhibitor of embryonic folate metabolism [13], histone deacetylases (HDACs) [14], and glycogen synthase kinase-3 (GSK3) [15], and by increasing oxidative stress [16]. However, the precise teratogenic mechanism(s) of VPA remains unknown.

Open NTDs are also known to have a large genetic risk theoretically due to rare de novo or inherited deleterious genetic variants. This genetic risk is shown by a higher concordance of NTD for monozygotic twins (7.7%) than dizygotic twins (4.0%) [17,18]. Additionally, after a single NTD pregnancy, the risk of recurrence rises from ~1/1000 to 1/20. Despite this accumulating evidence for a large genetic risk component and >350 gene genes known to cause NTDs in knockout mice, little is known about the genetic variants that cause NTDs in humans. This is likely due to complex genetics, incomplete penetrance, gene–environment interactions (e.g., folic acid), and lack of information on the effects of specific mutations (e.g., missense) [19]. *SHROOM3* loss-of-function variants have been shown to cause exencephaly in genetic mouse models and have been identified in human open NTD pregnancies [20,21,22,23,24]. SHROOM3 is an important scaffold protein for apical–basal cell polarity organization [25]. It coordinates apical constriction via polarized recruitment of rho-kinase, which leads to phosphorylation of non-muscle myosin and, ultimately, constriction of the apical actomyosin cytoskeletal network. This polarized apical constriction is necessary to initiate the process of neurulation. In addition to rare or de novo germline mutations, a recent hypothesis for the sporadic nature and lack of genetic variant identification in many human NTD pregnancies is that somatic mosaic mutations in the neural tube can also lead to an open NTD. One recent study provided evidence for this hypothesis by showing that when only 16% of cells in the mouse embryo neural tube had the NTD gene *VANGL2* knocked out, spina bifida resulted [26]. Identification of potential human NTD variants and characterization of these variants in robust human NTD models is needed to provide information for effective genetic counseling and family planning.

Human pluripotent stem cell-based models already provide greater predictive value than animal models for certain types of toxicological screening. For example, proarrhythmic risk is now routinely assessed in iPSC-derived cardiomyocytes [27]. Not only can these human models avoid false negative results, such as with thalidomide, but they may potentially limit false positives, allowing for a larger number of medications to enter clinical trials. With the advent of 3D brain organoid technology, several groups have made attempts to model neuroteratogenicity [28]. Unfortunately, standard methods result in highly variable organoids with multiple rosettes [28]. Since neural rosettes are the in vitro correlate of the developing neural tube, a multi-rosette model does not recapitulate normal human brain development adequately for detailed structural analyses. Therefore, most models have used a transcriptomic approach to assess the neuroteratogenicity of compounds, such as the STOP-Tox_UKN_ system [29], or involved complicated machine learning algorithms to identify and measure 2-dimensional structures in the neural rosette formation assay [30]. Attempts have also been made to model genetic NTDs, as well [31,32]. One study compared brain organoids from spina bifida aperta patients with controls finding differences in the size and number of neural rosettes between the disease groups [32]. However, this study lacked isogenic controls, and iPSCs were generated from different tissue types. A more recent study modeled recurrent anencephaly due to recessive mutations in *NUAK2* with iPSCs and organoids. While reaching statistical significance, the structural organoid measurements were extremely heterogeneous due to multiple rosettes and a variable sectioning plane. Furthermore, the structural results were difficult to correlate with the mechanism [33]. Given these issues, a 3D brain organoid model that recapitulates neurodevelopment in humans with higher fidelity would be useful to examine the genetic mechanisms of NTDs.

Our group and others have published techniques for generating single rosette organoids or neural cyst cultures [34,35,36,37,38]. Our simple model of human neurulation, which we call SOSR-COs (self-organizing single-rosette cortical organoids), overcomes the lack of disease-relevant structural readout in previous brain organoid models. Herein, we demonstrate clear, in vitro, dose-responsive structural phenotypes for acute VPA treatment, characterize the transcriptomic effects of VPA, and identify inhibition of apical constriction via HDAC inhibition as the likely mechanism by which VPA causes NTDs. Furthermore, we demonstrate an altered structure, polarization, and reduced apical constriction for *SHROOM3* knockout SOSR-COs, and we find that these phenotypes are gene-dose responsive in mosaic mixing experiments. These findings provide additional evidence that somatic mosaic variants could lead to NTD formation.

## 2. Materials and Methods

### 2.1. iPSC Lines

We purchased the AICS-0023 cell line from the Coriell Institute Biorepository (Camden, NJ, USA) in the Allen Cell collection. This line contains a monoallelic mEGFP-TJP1 (which encodes for the zona occludens-1 [ZO1] protein) to label the lumen in live SOSR-COs. The *SHROOM3* knockout line and isogenic control lines were generated by simultaneously utilizing CRISPR indel formation and an iPS cellular reprogramming technique we previously published [39]. In short, commercially available foreskin fibroblasts were nucleofected with episomal reprogramming vectors [40] and the pX330 CRISPR plasmid (Addgene, Watertown, MA) containing a gRNA targeting the 5th exon of the *SHROOM3* gene near a premature stop codon identified in an anencephalic human fetus [22]. Clonal iPSC lines generated by the reprogramming were PCR amplified over the gRNA cut site and Sanger sequenced. One isogenic control and one compound heterozygous frameshift (−1/−14 bp) clone were identified and sequenced by NGS to ensure a 1:1 ratio of the two mutant alleles. The gRNA and PCR primer sequences are listed in Appendix A.

### 2.2. iPSC Culture

The ips cell cultures were maintained on 6-well tc dishes coated with geltrex (Thermo, Waltham, Ma, USA) diluted 1:200 dilution in dmem/f12 medium (Thermo). Cells were cultured in mtesr1 medium (Stemcell Technologies, Vancouver, BC, Canada). When the colonies reached ~40% confluency, the cultures were incubated with 1 ml of hypertonic solution containing sodium citrate and potassium chloride for 2 min [41]. The solution was then replaced with mtesr1, and the colonies were mechanically detached with a mini cell scraper. The solution was then pipetted up and down 3–6 times to break the colonies into smaller pieces. The solution was then re-plated at a dilution of 1:8 onto newly geltrex-coated plates.

### 2.3. SOSR-COS Differentiation

We Accutase passaged 7–8 × 10^5^ iPSC lines onto Geltrex-coated (1:50 dilution in DMEM/F12) 12-well plates in mTeSR1 with 10 µM Y-27632 (Tocris, Bristol, UK). When the cells reached 80–100% confluency, the medium was changed to 3N (50:50 DMEM/F12:neurobasal with N2 and B27 supplements [Thermo]) without vitamin A with 2 µM DMH-1 (Tocris), 2 µM XAV939 (Cayman Chemical, Ann Arbor, MI, USA), and 10 µM SB431542 (Cayman Chemical) with 2 mL of medium per well. Subsequently, 1.5 mL medium were changed daily with 1 µM cyclopamine (Cayman Chemical) added beginning on day 1. On day 4, the monolayer was cut into squares using the StemPro EZ passage tool (Thermo). The squares were incubated for 2 min with the hypertonic sodium citrate solution. After aspiration, the squares were sprayed off the bottom with 1 mL of preconditioned culture media with a P1000 micropipette. An additional 1 mL of fresh culture medium with the 4 inhibitors was added. Approximately 100 µL of resuspended monolayer squares were then transferred into the wells of black-walled thin-bottom 96-well plates (µClear [Greiner Bio-One, Monroe, NC, USA] or Phenoplate [PerkinElmer, Waltham, MA, USA]) preincubated at 37 °C for 30 min with 35 µL of 100% Geltrex solution in each well.

### 2.4. SOSR-CO Drug Treatment

Immediately after plating in 96-well plates, the SOSR-COs were treated with various drugs for 48 h. Along with VPA (Cayman Chemical), the SOSR-COs were treated with different HDAC inhibitors, GSK3β inhibitors, folic acid (FA) inhibition or supplementation, and antioxidants. We tested HDAC inhibitors that target different classes: Trichostatin A (Cayman Chemical), Nicotinamide (Cayman Chemical), and CI-994 (Cayman Chemical). The GSK3β inhibitors included CHIR99021 (Cayman Chemical), BIO (Cayman Chemical), SB-216763 (Cayman Chemical), SB-415286 (Cayman Chemical), and lithium chloride (Sigma-Aldrich, St. Louis, MO, USA). FA (Cayman Chemical) was supplemented, and the FA pathway was inhibited by aminopterin (Cayman Chemical). The antioxidant was vitamin E (±-α-Tocopherol Acetate; Cayman Chemical). Appropriate amounts of each treatment were mixed with 3N media without vitamin A, plus the 4 inhibitors. A total of 100 μL of media mixed with the treatment was added per well.

### 2.5. SOSR-CO Fixing & Staining

The SOSR-COs were fixed at day 6, 48 h after drug treatment, with 4% paraformaldehyde in phosphate buffered saline (PBS) for 1 h at 4 °C. Following the fixation, the cells were washed with PBS twice for 5 min at room temperature. The SOSR-COs were permeabilized in PBS with 0.1% TritonX-100 for 30 min, followed by a 1-h incubation in ICC blocking buffer (PBS with 0.05% Tween-20, 5% normal goat serum, and 1% BSA) + 0.1% TritonX-100. The primary antibodies were mixed with this same blocking buffer and incubated with the SOSR-COs overnight at 4 °C. See Appendix A for the antibody catalog numbers and dilutions. The SOSR-COs were then washed 3 times with PBS containing 0.1% Tween-20 (PBST). The secondary antibodies, as well as phalloidin-Alexa488 conjugate (AAT Bioquest, Pleasanton, CA, USA) were added to the ICC blocking buffer and incubated with the SOSR-COs overnight at 4 °C. The cells were washed 1 time with PBST. The SOSR-COs were then counterstained by incubation with either bis-benzimide or HCS CellMask DeepRed (Thermo) for 1 h at room temperature, followed by 3 additional PBST washes. Alternatively, when immunolabeling was not necessary during the drug treatment experiments, the SOSR-COs were counterstained directly following fixation and then washed with PBS that did not contain Tween-20. In this case, the constitutive expression of the ZO1-EGFP fusion protein was used for image analysis.

### 2.6. Manual SOSR-CO Imaging and Analysis

Images were obtained on an EVOS Cell Imaging System (Thermo). Four images of different SOSR-COs were obtained per well. We excluded SOSR-COs that were smaller than 100 microns or greater than 200 microns in diameter. Larger SOSR-COs tended to be fusions from 2 monolayer fragments, while smaller SOSR-COs were from smaller pieces of the monolayer. Each SOSR-CO was imaged for ZO1-EGFP along with bis-benzamide, creating 8 images in total for each well (2 per each SOSR-CO × 4 SOSR-CO per well). ImageJ was used to set the maximum and minimum thresholds to optimize the contrast between the lumen (ZO1-EGFP or ZO1 immunostaining) or SOSR-COs (bis-benzamide for DNA) with the surrounding background before creating a binary mask. After using the “fill holes” function, the area, circularity, roundness, solidity, and aspect ratio for each image was obtained.

### 2.7. Automated Imaging and Analysis

Automated confocal microscopy was utilized to increase the throughput. For confocal microscopy, the PBS solution was removed from each well and a fructose glycerol solution was added. This solution increased the SOSR-CO clarity by correcting the refractive index and dehydrating the Geltrex hydrogel, causing the SOSR-COs to move to the bottom of the dish and allowing one imaging range in the Z-axis across the plate. For imaging, we used the Cell Voyager 8000 (Yokogawa, Tokyo, Japan) automated spinning disc confocal microscope with a 10× dry (NA = 0.40) UPLSAPO10X2 (Olympus, Tokyo, Japan) objective lens. For each well of the 96-well plates, 9 fields were imaged with 16 Z-sections with 6 μm spacing and a 100 μm total depth. Laser autofocus was performed by undersurface detection and offset of 90 μm into the cell plane. This resulted in imaging the vast majority of SOSR-COs in each well. The optical sections were used to generate a maximum intensity projection for each field. These images were imported into CellProfiler with a custom pipeline to identify the margin of each SOSR-CO (utilizing HCS CellMask Deep Red) and the apical lumen (utilizing ZO1-EGFP, ZO1 immunostaining, or phalloidin-Alexa488). Using these masks, the intensity and morphometric/shape features were measured, resulting in over 400 features (Appendix A). The database file was then imported into a custom Python script to join the experimental metadata (i.e., drug, concentration, genotype, and cell line). The script then determines the differences between the conditions in the number of lumens in each SOSR-CO and the differences in the SOSR-CO areas. In control lines under normal media conditions, multi-lumen SOSR-COs and max project areas > 31,415 µm^2^ are mostly driven by two monolayer fragments fusing. The analysis was restricted to single lumen SOSR-COs with areas between 7854–31,415 µm^2^. This corresponds to the 100–200 µm diameter criterion used for manual imaging. A custom analysis pipeline was coded in Python using Jupyter notebooks which can be found here: https://github.com/huangjoy1/sosr-analysis-files.

### 2.8. Random Forest Predictive Modeling

To identify the most informative features for discriminating between conditions (VPA concentration or *SHROOM3* genotype), we first scaled and centered each metric. We then utilized the bootstrap random forest model and calculated variable importance metrics with standard 5-fold cross validation. This modeling was performed on JMP Pro 16.

### 2.9. RNA Sequencing Experiment

Day 6 brain organoids were generated from a male iPSC line (AICS-0023) and treated for 48 h with either a vehicle or one of the following 3 compounds: (1) the non-teratogenic compound trichostatin-A; (2) the teratogenic anti-seizure medication VPA at a concentration of either 200 µM or 400 µM; or (3) CHIR99021 at a concentration of 10 uM. All the treatments were performed in 4 replicates, except for CHIR99021, which was performed in triplicate. The total RNA was isolated using the RNeasy PLUS Universal Kit (Qiagen, Germantown, MD, USA). The SOSR-COs were solubilized along with the Geltrex hydrogel using Qiazol, and the kit was used according to the manufacturer’s recommendations. Libraries were constructed using PolyA RNA ultra II (New England Biolabs, Ipswich, MA, USA) and subsequently subjected to 150 cycles of sequencing on NovaSeq-6000 (Illumina, San Diego, CA, USA). The adapters were trimmed using Cutadapt (v2.3). FastQC (v0.11.8) was used to ensure the quality of the data. The reads were mapped to the reference genome GRCh38 (ENSEMBL)] using STAR (v2.6.1b) and assigned count estimates to genes with RSEM (v1.3.1). The alignment options followed ENCODE standards for RNA-seq [4]. FastQC was used in an additional post-alignment step to ensure that only high-quality data were used for expression quantitation and differential expression. The data were pre-filtered to remove genes with 0 counts in all the samples. A differential gene expression analysis was performed using DESeq2, using a negative binomial generalized linear model (thresholds: linear fold change >1.5 or <−1.5, Benjamini-Hochberg FDR (Padj) < 0.05). The genes were annotated with NCBI Entrez GeneIDs and text descriptions. Plots were generated using variations of DESeq2 plotting functions and other packages with R (version 3.3.3). The data for all expressed genes comparing between the treatment and vehicle controls are given in Appendix A. The RNA sequencing data files in this publication have been deposited in NCBI’s Gene Expression Omnibus and are accessible through GEO Series accession number GSE232218 (https://www.ncbi.nlm.nih.gov/geo/query/acc.cgi?acc=GSE232218 (accessed on 18 May 2023)).

### 2.10. Quantifying Apical Surface Areas

SOSR-COs counterstained and/or immunostained as described above were imaged with an oil 100× objective on a Stellaris 5 inverted confocal microscope (Leica, Wetzlar, Germany). The Z-series were set at immediately above and below each lumen unless limited by working distance. A step size of 0.75 µm was used. The images were deconvoluted with the Lightning program from Leica. The image stacks were then imported into FIJI, where sections were used to make a maximum Z projection of the hemisphere above the Geltrex hydrogel. A pipeline in CellProfiler 2 was then used to accentuate the ZO1 labeled tight junctions. Approximately 30 apical surface areas were measured by hand in FIJI for each maximum projection, avoiding the edges. We sought to measure all of the surface areas in the center of the ZO1-labelled lumen to avoid size bias or surfaces orthogonal to the imaging plane.

## 3. Results

### 3.1. High Content Imaging Pipeline for SOSR-CO NTD Screening

We have previously reported a protocol for generating self-organizing single rosette spheroids with manual imaging and analysis [37]. To dramatically increase the throughput and decrease human bias, we have developed an entirely automated pipeline for imaging and analysis. First, we generated the neuroepithelial monolayer by four inhibitor treatment of iPS cells for 4 days. At this time, we cut the monolayer into identically sized fragments and added the fragment suspension into a 96-well plate coated with Geltrex hydrogels (Figure 1A). After 48 h of exposure, the cultures were paraformaldehyde fixed and stained. We then utilized a Yokogawa CV8000 automated dual microlens confocal to generate 10× images of each well of the 96-well plate containing on average 20–40 SOSR-COs per well (Figure 1B). The confocal optical sections were compressed into a maximum projection image (Figure 1C). Utilizing a CellProfiler pipeline, each SOSR-CO was identified from these images, as well as the central lumen labeled with either ZO1-EGFP fusion protein, ZO1 antibody, or phalloidin for labeling f-actin (Figure 1D). Over 400 features, including shape, fluorescence intensity, radial distribution, and area metrics, were extracted from the images (Figure 1E). The SOSR-CO datasets were then filtered for individual lumens and size (7800–31,415 µm^2^). We used these filters to exclude SOSR-COs that resulted from the fusion of multiple monolayer fragments, as shown in our previous preprint manuscript [37]. To identify the distinguishing features of particular treatments or genetic variants, we utilized the random forest predictive model, followed by inspecting the individual feature importance. For pharmaceutical treatments, linear regression random forest decreases the risk of false feature identification. The distinguishing features were then validated by univariate comparisons and statistical testing.

### 3.2. VPA Dose Response Shows SOSR-CO Lumen Size as Most Defining Feature

VPA, a well-known cause of NTDs in mice and humans, previously gave consistent expansion of the lumen area normalized to the entire SOSR-COs area at 200 and 400 µM doses [37]. Here, we performed a larger dose-response curve from 50 to 800 µM using our newly established pipeline. Our previously engineered feature, the normalized lumen area, showed consistent increases across the entire dose-response curve (Figure 2a,b). With the increased statistical power from the large number of SOSR-COs imaged by the automated confocal, we noted a significant difference between any two concentrations by one-way ANOVA with a multiple comparison post hoc test. The same relationship can be seen from our original manually collected data across three independent experiments in Appendix A. Using a linear regression-based bootstrap random forest predictive model (Appendix A) with the entire dataset (>400 extracted features), we found a high degree of predictiveness (R^2^ = 0.945) (Appendix A). This model held true for the standard fivefold cross validation (R^2^ = 0.811) (Appendix A). The normalized lumen area (area ratio) was the most instructive feature for identifying the group with 30% of the overall impact (Appendix A). All the other top instructive features are almost entirely a radial distribution of the ZO1-EGFP fluorescence-based features collectively making up 41.5% of the overall impact. The radial distribution proved to be useful in cases where the lumens were extremely large or had dysmorphic shapes, as it did not require outlining the lumen, but instead relied on the fluorescent label of the lumen within the SOSR-CO mask. The radial distribution utilizes six concentric shapes within the SOSR-CO with an identical diameter and the amount of EGFP signal is quantified in each of these rings (Figure 2c). We normalized the data at each concentration to the most intense ring to account for the reduced signal intensity as the lumen expands. We then generated a three-color heatmap for plots of individual concentration averages and the entire dose response curve (Figure 2c,d). The gradual shift in fluorescence primarily from the inner three rings to the outer three rings is apparent. To quantitatively represent these same data, we have plotted the mean fraction of the EGFP signal for rings 2–4, which showed the most robust VPA-dependent changes. Ring 2 decreased from 200 to 800 µM, ring 3 increased from 50–400 µM, and ring 4 increased from 200–800 µM VPA (Figure 2e–g). These data demonstrate the progressive increase in average lumen size with an increased dosage of VPA.

### 3.3. Lumen Enlargement Is Directly Related to Reduced Apical Constriction

Since apical constriction is a necessary mechanism of neurulation and inhibitors of apical constriction lead to NTDs in mouse models and increased lumen size in our SOSR-CO model [37], we decided to measure the apical surface areas of SOSR-COs treated with the dose curve of VPA. We utilized ZO1 immunostaining to increase the fluorescent signal for high magnification confocal microscopy (100×). This tight junction marker outlines each cell at the apical surface, and these outlines were measured to determine the area of each apical surface. We observed a dramatic increase in the apical surface areas with increasing doses of valproic acid (Figure 2h). These results were significant when compared with the vehicle starting at 100 µM VPA and increasing up to 800 µM (Figure 2i). We then compared the relationship between the lumen size and apical surface area size by plotting the means of each at 0, 100, 200, 400, and 800 µM VPA on an XY scatter plot (Figure 2j). A linear regression analysis showed a striking level of correlation between these two measurements with an R^2^ of 0.994. Therefore, we believe measurement of the lumen area is a reasonable proxy measurement for the cell apical surface areas in a given SOSR-CO. This is expected because the 3D lumen surface area is the sum of all of the apical surface areas; the lumen must have a larger size to accommodate the larger individual surfaces.

### 3.4. GSK3β or HDAC Inhibition Cause VPA-like Phenotypes, but No Evidence for Folic Acid or Oxidative Stress Involvement

With the clear phenotypic characteristics of the VPA-treated SOSR-COs, we attempted to find compounds that would ameliorate or mimic the effects of VPA on the lumen size. First, we attempted to block the effects of VPA with folic acid supplementation, the most effective prophylactic treatment for NTDs and a possible VPA target [13]. We also tested whether aminopterin, a potent inhibitor of the folic acid pathway, could have an effect similar to VPA. We found no effect of 10 µM folic acid on the normalized lumen size at 0, 200, or 400 µM VPA (Figure 3a). Aminopterin exposure did not affect the lumen size (Figure 3b); thus, the evidence suggests that VPA does not alter the lumen structure and apical constriction via inhibition of the folic acid pathway. One caveat is that our organoid cultures may be resistant to aminopterin due to an abundance of hypoxanthine and thymidine in the growth medium that can be used to generate purines through the salvage pathway without needing folic acid metabolism. Amino acids synthesized from the folic acid pathway precursors are also abundant (i.e., glycine, serine, methionine). Thus, the major pathogenic mechanisms of folic acid deficiency are likely avoided.

To test the hypothesis that VPA caused NTDs by increasing the oxidative stress [16], we added antioxidant, vitamin E, during VPA exposure. Again, there was no effect of either 10 or 100 µM vitamin E on the lumen size for the vehicle or 400 µM VPA treatment (Figure 3c). To reiterate, these data suggest that increased oxidative stress is not the mechanism by which VPA causes decreased apical constriction. The most well-known action of VPA is HDAC inhibition [14]. Treatment with HDAC I and HDAC III inhibitors, nicotinamide and CI994, respectively, did not affect the lumen size, but trichostatin A (TSA), an HDAC I/II inhibitor (as is VPA), caused a similar dose-response increase in the lumen size (Figure 3d,e). High magnification microscopy also showed that 100 nM TSA causes dramatic enlargement of the apical surfaces areas compared with the vehicle and to the same extent as 800 µM VPA (Appendix A). VPA is also known to inhibit GSK3β [15]. Inhibitors of GSK3β, including CHIR99021, CHIR98014, SB216763, SB415286, BIO, and lithium, all caused increased lumen size (Figure 3f–i). Exemplary images showing the increased lumen size are shown in Appendix A. Therefore, both HDAC and GSK3β inhibition decrease apical constriction, and one or both are possible mechanisms by which VPA causes NTDs.

### 3.5. RNA Sequencing Results Suggest VPA Inhibits HDAC but Not GSK3β at Concentrations Causing NTD Phenotypes

Since HDACs and GSK3β signaling both have large transcriptomic effects, we characterized the transcriptomic changes caused by concentrations of VPA (at 200 and 400 µM) that inhibited apical constriction. We also compared the effects of a specific GSK3β inhibitor, CHIR99021 (10 µM), that also led to lumen size expansion. Four independent experimental sample sets were generated by extracting RNA from SOSR-COs after 48 h of drug exposure. A principal component analysis showed excellent clustering by treatment in PC2, while a single experimental replicate was highly divergent in PC1 (Figure 4a). At 200 µM, VPA-exposed SOSR-Cos had 800 differentially expressed genes (DEG => 50% change and a padj of <0.05) and 1833 at 400 µM with 783 shared DEGs. CHIR99021 exposure had 699 DEGs with only 70 and 170 DEGs shared with 200 and 400 µM VPA, respectively (Figure 4b). Despite massively altering the transcriptome, the most striking effect of VPA was an increase in the expression of neuronal genes. The top cellular compartment GO terms for increased VPA DEGs were nearly all neuronal, including “axon”, “neuronal cell body”, “postsynaptic membrane”, “synapse”, and “dendrite” (Figure 4c), and the top increased DEG for both 200 and 400 µM was *STMN2*, a quintessential transcriptional marker of neurons (Figure 4d,e). This effect on neuronal genes has been previously shown in neural progenitors due to increased acetylation of H4, which associates with the proneuronal *Ngn1* promoter [42]. Inhibition of GSK3β via CHIR99021 had many different top transcripts from VPA, although *STMN2* was still increased but to a lesser extent (Figure 4f). Inhibition of GSK3β mimics the effects of WNT signaling and should result in increased expression of TCF/LEF target transcripts. Indeed, targets *TCF7, DKK1, AXIN2,* and *CCND2* were all significantly increased with CHIR exposure (Figure 4g); however, these transcripts were not significantly affected by 200 or 400 µM VPA. Furthermore, a GO analysis of the DEGs showed that WNT was a significantly affected pathway by CHIR, but not VPA (Figure 4h–j; the yellow circle indicates the WNT pathway). These data show that VPA does not inhibit GSK3β activity at these concentrations, thus GSK3β is not the cause of the increased lumen area in VPA-treated SOSR-COs. It is more likely that the massive effect on the transcriptome due to HDAC inhibition causes the structural changes. Lastly, genes important for neurulation that are associated with NTDs (e.g., *SHROOM3*, *NUAK2*, *VANGL2, CELSR2*) were all decreased by VPA and CHIR99021 exposure (Figure 4k). The data for all the expressed genes comparing between treatment and vehicle controls are displayed in Appendix A.

### 3.6. SHROOM3 Knockout Results in Enlarged, Dysmorphic Lumens and Reduced Apical Constriction

To further investigate the utility of our NTD model system, we sought to generate a genetic model of NTDs. *SHROOM3* knockout mice have exencephaly, and several human anencephaly pregnancies with *SHROOM3* variants have been reported [22]. We therefore utilized our previously developed methodology for generating knockout iPSC lines by simultaneous reprogramming and CRISPR indel formation to generate a *SHROOM3* knockout iPSC line and isogenic control [39]. We used a gRNA in the same exon as a human anencephaly pregnancy with a premature stop codon mutation (Figure 5a) [22]. We generated a line with compound heterozygous frameshift mutations via +1/−14 bp indels at the CRISPR/Cas9 cut site, as well as an isogenic control without indel formation (Figure 5b–d). Utilizing quantitative reverse transcriptase PCR, we found that the *SHROOM3* transcript levels peaked on day 4 of differentiation at the same time that *PAX6* transcript also peaked (Appendix A). This fits with prior studies that found SHROOM3 expression is dependent on PAX6 expression [43]. A qRT-PCR did not show any difference between the isogenic control and our engineered knockout line, indicating a lack of nonsense-mediated decay (Appendix A). An immunoblot analysis had too many non-specific bands to confirm knockdown; however, using immunocytochemistry for SHROOM3 on whole mount SOSR-COs, we noted a ring of apical SHROOM3 expression in the isogenic control that is absent in the knockout line (Appendix A).

While our data do not conclusively show loss of *SHROOM3* expression, as you will see, the dramatic structural phenotypes in the lumens of our *SHROOM3* SOSR-COs fits with prior *SHROOM3*-KO mouse models. Instead of a small circular lumen, as seen in the isogenic controls (Figure 5e,l and Appendix A), we observed massively enlarged lumens with lobes protruding into the lumen space or occasionally into large openings at the top of the lumen, suggesting incomplete closure (Figure 5f,j and Appendix A). Both f-actin staining via a phalloidin-fluorophore conjugate (Figure 5e,f and Appendix A) and ZO1 immunostaining (Figure 5i,j) showed enlarged lumen phenotypes. We found radial distribution of both phalloidin stained f-actin and immunostained ZO1 were the most distinguishing factors for the genotype as compared WT to KO. The mean fraction of the fluorescence in ring 2 was the most informative with f-actin staining across 4 independent experiments (Figure 5g). The mean fraction of fluorescence in ring 4 was most informative with ZO1 staining across 4 independent experiments (Figure 5k). We also compared the distribution in ring 6. For ZO1 immunostaining, we did not distinguish any significant difference, demonstrating that the lumen tight junctions do not enter this extreme radius of the SOSR-COs (Figure 5l); however, f-actin staining did increase in ring 6, suggesting a disruption of the apical–basal polarization of f-actin (Figure 5h).

Again, our previous work indicates that lumen enlargement is due to reduced apical constriction. To test this hypothesis for *SHROOM3* knockout, we performed high-resolution microscopy and area measurements for the apical surface areas as defined by the tight junction marker, ZO1. The large apical surface areas can be seen by ZO1 immunostaining in *SHROOM3*-KO compared with the isogenic control (Figure 5m,n).The mean size increased greater than threefold from 2.5 to 7.7 µm^2^ (Figure 5o). These data fit with the literature that *SHROOM3* is necessary for polarized localization of apical constriction machinery and subsequent constriction. Therefore, reduced apical constriction is expected, as is the increased basal f-actin. Both have been seen as a consequence of *SHROOM3* loss of function in a mouse model [43].

### 3.7. Mosaic SHROOM3 Knockout Shows Gene-Dose Responsive Lumen Size

It is theorized that NTDs can be caused not only by germline mutations, but also somatic mutations in the neural tube, resulting in a mosaic loss-of-function of important proteins. One recent publication found that knocking out *VANGL2* in only 16% of the cells in the neural tube of a developing mouse embryo was sufficient to produce spina bifida [26]. Since we have both *SHROOM3*-KO and isogenic control lines, we sought to investigate this hypothesis by mixing the two genotypes in five different ratios (% *SHROOM3*-KO: 0, 12.5, 25, 50, 75, and 100), shown schematically in Figure 6a. We isolated the genomic DNA taken from one batch of mixed cultures at the day 4 monolayer stage and performed PCR across the indel site, followed by next-gen sequencing. We plotted the actual % of KO cells over the expected amount and found an R^2^ of 0.9998 (Appendix A). Similar to the VPA dose-response, we graphed the representative images and gene-dose effects on radial distribution of f-actin and ZO1 using a heatmap (Figure 6b–e). Using the random forest model on the image datasets from four independent experiments, we were able to train to a model with high correlation between the actual % of KO cells and predicted for either f-actin data (R^2^ = 0.939 for training set and 0.832 for validation set) or ZO1 data (R^2^ = 0.892 for training set and 0.517 for validation set) (Appendix A). We found that the same features (radial ZO1 and f-actin distribution) that best distinguished the 0 and 100% were the most distinguishing features across the mixed cultures (Appendix A). These data were plotted for either all the SOSR-Cos or the independent experiment means (Figure 6f–i). Interestingly, 12.5% KO was significantly different from 0% for each measurement, but 75% and 100% were indistinguishable from one another for each measurement (Figure 6f,h). These data clearly show the lumen size increases in direct relationship with the increased percentage of *SHROOM3*-KO cells.

## 4. Discussion

Our results show that our model system is useful for rapid screening of pharmaceuticals and genetic variants for NTD-like structural phenotypes. We have shown dose-responsive increases in the lumen size for VPA that correlates with decreased apical constriction. We have also shown that VPA has this effect primarily through HDAC inhibition. In this report, we also generated a genetic anencephaly model by knocking out *SHROOM3,* which resulted in increased lumen size and reduced apical constriction, similar to the effects of VPA. Additionally, the *SHROOM3* knockout also altered the polarity of filamentous actin. The effects of *SHROOM3* knockout were also shown to be gene-dose responsive in our mosaic mixing experiments, and these results may indicate that an open NTD could result when only a minority of cells in the neural tube contain a pathogenic variant. However, correlating subtle in vitro features to in vivo pathological rates is difficult to conject. Taken together, these data suggest that reduced apical constriction is a shared mechanism for these two quintessential causes of NTDs. Furthermore, these results, along with the robust high-throughput platform we have developed, and the statistical power of the model system suggest that our system would perform well for screening for novel pharmaceutical or genetic causes of NTDs.

Inhibitors of apical constriction have been shown previously in our system and in rodent embryo models to result in enlarged dysmorphic lumens and NTDs, respectively [37,44]. *SHROOM3* has been shown to be necessary and sufficient to induce apical constriction in cell culture and embryo models [45], but the mechanism by which VPA leads to NTDs has been less clear. While our data indicate that VPA causes NTDs via HDAC inhibition, which of the 800+ genes with altered expression and what particular functions are impinged are not clear from the transcriptomic data alone. Our structural phenotyping clearly shows that apical constriction is reduced by VPA. VPA also led to a decrease in the expression of 4-NTD related genes, including *SHROOM3*, but further study is needed to understand which transcriptional changes lead to the outcome of reduced apical constriction. Our data differ dramatically from one recently published study, which found 1 µM VPA was sufficient to disrupt single rosette formation by inhibiting apical f-actin and that this effect could be blocked by the addition of 10 µM folic acid [36]. Our study shows that 50 µM VPA is needed to see clear evidence of lumen enlargement and that 10 µM folic acid had no effect on the structural changes seen with 200 or 400 µM VPA. One key difference is that the former published study treated cultures with VPA starting at day 0 of neural induction, while our study added VPA when neurulation was induced on day 4. Our transcriptomic data showed a modest but dose-response decrease in *SHROOM3* expression with VPA treatment. The loss of apically polarized f-actin but not ZO1 in this prior study is strikingly similar to our *SHROOM3* knockout findings [36]. Thus, early administration of VPA may inhibit *SHROOM3* expression, resulting in mislocalized f-actin. However, the differences in folic acid rescue between this study and ours would still not be explained unless folic acid somehow induces *SHROOM3* expression.

We also unexpectedly found that inhibition of GSK3β results in the same enlargement of the apical lumen and presumably decreased apical constriction. Our data also show that lithium exposure, which has controversially been linked to NTDs, could be a cause at high concentrations due to inhibition of GSK3β. The mechanism is likely due to alterations in β-catenin signaling and downstream transcriptional alterations. Interestingly, CHIR99021 exposure in SOSR-COs also decreases *SHROOM3* expression similarly to VPA. Therefore, inhibition of *SHROOM3* expression leading to reduced apical constriction could be a shared mechanism for VPA and GSK3β inhibitors and should be further explored, possibly by rescue experiments. Combining RNA-sequencing data from multiple models may also allow the identification of NTD biomarkers. Biomarkers have already been identified in maternal serum for NTDs, including PCSK9 protein [46].

Like all model systems, the SOSR-CO model has limitations. While highly amenable to high-throughput techniques, there are still important aspects in whole embryos or multi-lineage systems that cannot be recapitulated. For instance, our current SOSR-CO methodology has no developmental axis patterning, such as dorsal–ventral or rostral–caudal. The spheroids are all patterned to be dorsal forebrain [37]. This is vitally important to the study of spina bifida, which occurs at the caudal rather than rostral extreme of the neural tube. Additionally, while many lines of data indicate spina bifida and anencephaly have many shared mechanisms, some causes seem to be specific to one or the other. Future work to develop a lumbar spinal cord patterned single-rosette CNS organoid would be useful in this regard.

In the future, the SOSR-CO model will need to be evaluated for predictive accuracy for a larger panel of known teratogenic and non-teratogenic compounds. If the predictive nature we see in the current study with VPA and *SHROOM3* continues for such panels, the SOSR-CO model may prove to be an ideal platform for identifying neuroteratogenic compounds during drug development or for existing compounds. These studies are vitally important for novel therapeutic screening and comparisons within drug classes known to carry NTD risk, such as antiseizure medications and possibly HIV antiretroviral therapies [10,47]. Additionally, our current results indicate that possible genetic NTD risk variants can be assessed in this system, but this warrants further investigation of outcomes in a larger number of well-characterized NTD risk genes, such as *VANGL2* and *SCRIB* [23,48].

## 5. Conclusions

Using our SOSR-COs system, we have developed a high-throughput image analysis pipeline for evaluating for NTD-like phenotypes. VPA increases the apical lumen size and apical cell surface area in a dose-responsive manner. Knockout of *SHROOM3*, a well-known NTD-related gene, also caused expansion of the lumen and increased the apical surface areas in a mosaic gene-dosage-dependent manner. The increased lumen sizes were caused by reduced cell apical constriction, suggesting that impingement of this process is a shared mechanism for VPA treatment and *SHROOM3*-KO, two well-known causes of NTDs. Our system allows the rapid identification of NTD-like phenotypes for both compounds and genetic variants and should prove useful for understanding specific NTD mechanisms and predicting drug teratogenicity.

## 6. Patents

The following patent resulted from this work: Parent, J. and Tidball, A. (2023). Stem cell-derived single rosette brain organoids and related uses thereof. U.S. Patent Application NO. 17/920.599.

## Figures and Tables

**Figure 1 cells-12-01697-f001:**
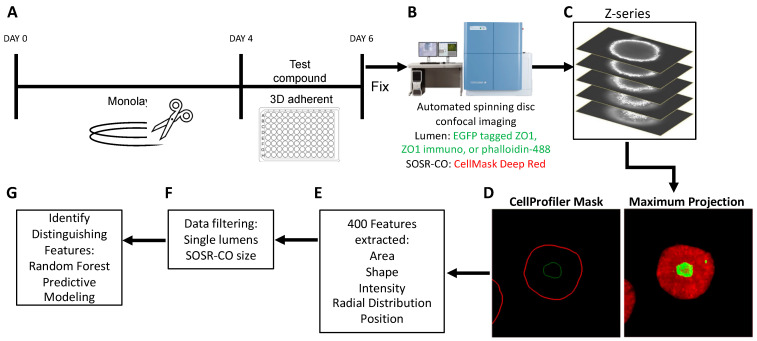
Automated platform for SOSR-CO NTD phenotyping. (**A**) Schematic of drug treatment paradigm during SOSR-CO neurulation. Monolayer neuroepithelial cultures were differentiated from iPS cells over 4 days. The monolayer was then cut with the StemPro EZ passage tool into reproducible fragments. These fragments were moved to a 96-well plate containing Geltrex and test compounds. Cells were cultured for 48 h, followed by paraformaldehyde fixation and counterstaining. (**B**) SOSR-COs were imaged for the constitutively tagged ZO1-EGFP and HCS CellMask Deep Red counterstain using the Yokogawa CellVoyager 8000. (**C**) Seven optical sections through the center of the SOSR-COs were generated. (**D**) The optical sections were used to generate a maximum projection in the far red and green channels. CellProfiler was used to identify the margins for each channel for each SOSR-CO. (**E**) CellProfiler extracted over 400 features from the SOSR-COs in the two channels, including area, shape, intensity, radial distribution, and position metrics. (**F**) Data were filtered for individual lumens and SOSR-CO area (7800–31,415 µm^2^). (**G**) Random forest model is used to identify distinguishing features based on treatment.

**Figure 2 cells-12-01697-f002:**
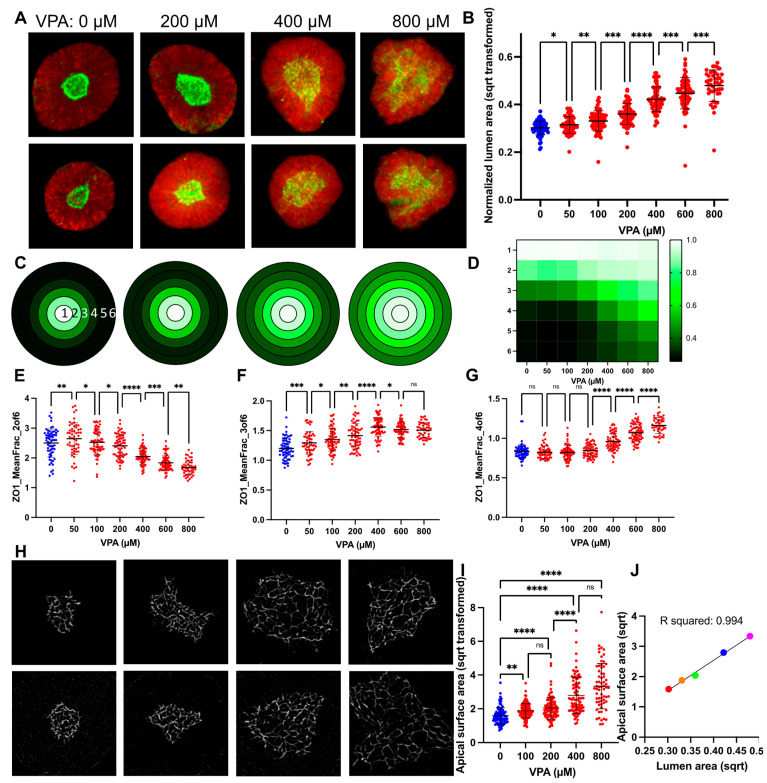
VPA causes a dose-response increase in SOSR-CO lumen size due to reduced apical constriction. (**A**) Example images of automated microscopy images of SOSR-COs treated with 0, 200, 600, and 1000 µM VPA. Green is ZO1-EGFP and red is HCS CellMask Deep Red. (**B**) SOSR-CO lumen size (normalized to total SOSR-CO area and square root transformed) is plotted for VPA doses from 50 to 1000 µM. Data from replicate independent experiments collected by manual imaging and analysis are presented in Appendix A. (**C**) Mean fraction of radial distribution plots (maximum normalized) for ZO1-EGFP fluorescence are depicted for the 4 concentrations in A. Each concentric ring represents the 6 concentric rings of equal thickness generated from each individual SOSR-CO and averaged for each group. Legend in D. (**D**) Heatmap of mean fraction radial distribution of ZO1-EGFP fluorescence across the entire VPA dose-response curve. Intensity is shown by 3 color heatmap, with white as the highest intensity, black as the lowest, and green as intermediate. (**E**–**G**) Quantitative data for radial distribution of EGFP-ZO1 in rings 2, 3, and 4 for the VPA data set. Each dot is an individual SOSR-CO. n = 63, 54, 66, 64, 68, 69, 45, and 32, respectively, for (**B**,**E**–**G**). (**H**) Example 100× confocal images of tight junctions on the apical lumen surface labeled by ZO1 immunostaining with increasing VPA concentrations (0, 200, 400, and 800 µM, respectively). (**I**) The size of individual cell apical surfaces as measured by the space outlined by the ZO1-EGFP-labelled tight junction is plotted. Kruskal–Wallis statistical test was performed with Dunn’s multiple comparison post hoc between groups. (**J**) XY scatter plot comparing the lumen max projection area to cell apical surface area means at each concentration of VPA. Colors are for increasing doses of VPA (red = 0, orange = 100, green = 200, blue = 400, and magenta = 800 µM VPA). Line and R2 value for linear regression analysis. Dose comparisons for (**B**,**E**–**G**) were performed by one-way ANOVA with FDR corrected post hoc test between groups. Bars are mean and standard deviation. * *p* < 0.05, ** *p* < 0.01, *** *p* < 0.001, **** *p* < 0.0001, ns *p* > 0.05.

**Figure 3 cells-12-01697-f003:**
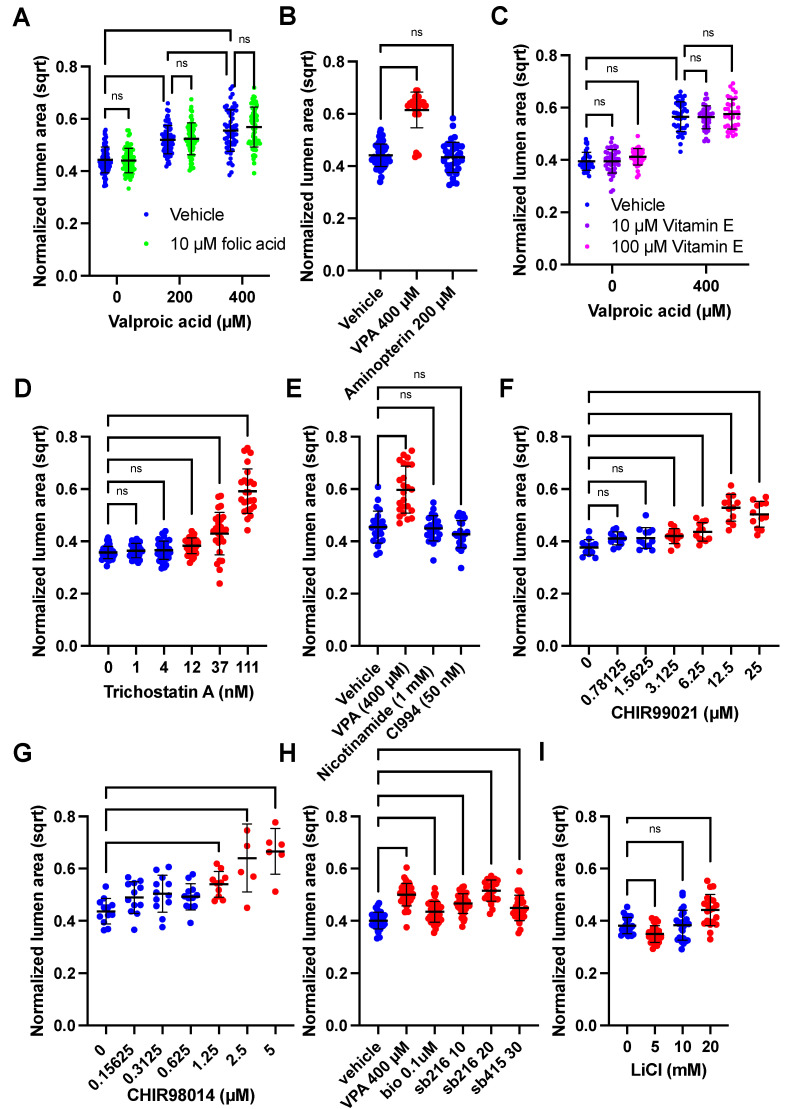
HDAC and GSK3β inhibitors increase SOSR-CO lumen size similar to VPA. (**A**–**H**) SOSR-COs were treated with various compounds with or without valproic acid, and the lumen area normalized to the total SOSR-CO area and square root transformed is plotted for each set of experiments. Each dot is an individual SOSR-CO. The number of total SOSR-COs and total independent experiments are listed below. Analysis was performed by manual microscopy and blinded semi-automated analysis. (**A**) SOSR-COs were treated with vehicle, 200, or 400 µM valproic acid with or without 10 µM folic acid, in addition to the basal media concentration (7.5 µM). n = 71, 71, 66, 63, 56, and 66, respectively, across 3 independent experiments. (**B**) SOSR-COs were treated with vehicle, 400 µM VPA, or 200 µM aminopterin (folic acid pathway [DHFR] inhibitor). n = 55, 34, and 37, respectively, across 2 independent experiments. (**C**) SOSR-COs were treated with vehicle or 400 µM VPA with the addition of 0, 10, or 100 µM vitamin E (antioxidant). n = 40, 40, 36, 40, 36, and 36, respectively, across 2 independent experiments. (**D**) SOSR-COs were treated with a dose curve of the class I/II HDAC inhibitor, trichostatin A. n = 40, 28, 39, 38, 28, and 24, respectively, across 3 independent experiments. (**E**) SOSR-COs were treated with HDAC inhibitors nicotinamide (1 mM) and CI994 (50 nM) and compared to 400 µM VPA. n = 24 for each group across 2 independent experiments. (**F**) SOSR-COs were treated with a log2 dose curve of the GSK3β inhibitor CHIR99021. n = 12 for each group for 1 experiment. (**G**) SOSR-COs were treated with a log2 dose curve of the GSK3β inhibitor CHIR98014. n = 12, 12, 12, 12, 10, 6, and 6, respectively, for 1 experiment. (**H**) SOSR-COs were treated with VPA and compared with other GSK3β inhibitors BIO, SB216763, and SB415286. n = 36, 36, 36, 36, 24, and 29, respectively, across 3 independent experiments. (**I**) SOSR-COs were treated with 0, 5, 10, and 20 mM lithium chloride. n = 24, 24, 24, and 20, respectively, across 2 independent experiments. Bars are mean and standard deviation. Statistical analyses were performed by two-way ANOVA with Tukey post-hoc test (**A**,**C**), Brown–Forsythe and Welch ANOVA (**D**,**E**) or one-way ANOVA with Dunnett’s multiple comparisons test (**B**,**F**–**I**). ns *p* > 0.05.

**Figure 4 cells-12-01697-f004:**
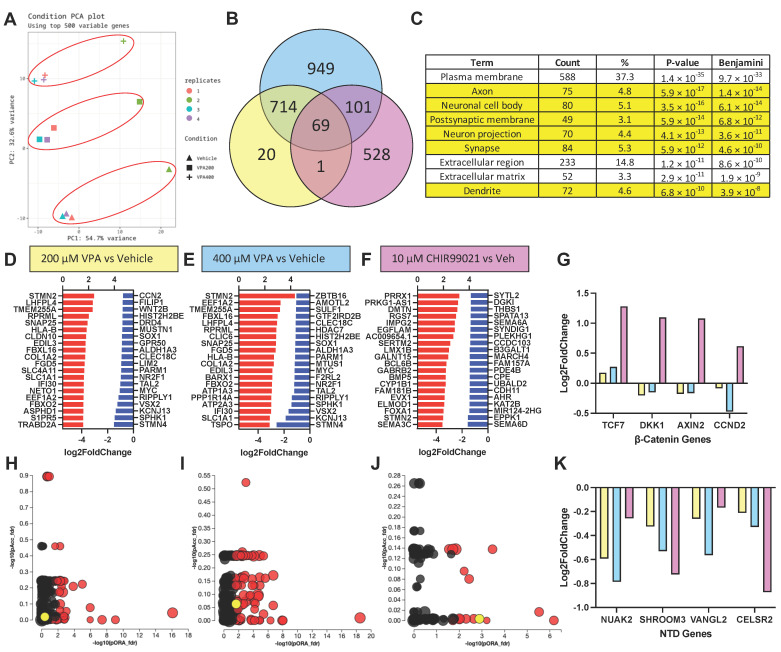
VPA causes transcriptomic signatures of neural differentiation, but not GSK3β inhibition. (**A**) Principal component analysis of VPA (200 and 400 µM) and vehicle-treated SOSR-CO mRNA sequencing analysis for 4 independent replicates. Red circles indicate treatment groupings. (**B**) Venn diagram of differentially expressed genes (DEGs) for 200 and 400 µM VPA and 10 µM CHIR99021 compared with vehicle control in SOSR-COs. Overlapping areas indicate the number of shared DEGs between treatment groups. (**C**) Gene ontology analysis of upregulated DEGs for 400 µM VPA treatment based on protein subcellular localization. Of the 9 top terms, 6 are neuron-specific (i.e., axon, dendrite, synapse). (**D**–**F**) The top 20 upregulated and downregulated transcripts for each treatment: (**D**) 200 µM VPA, (**E**) 400 µM VPA, and (**F**) 10 µM CHIR9901. (**G**) Log2 fold change in transcript levels for 4 canonical Wnt/β-catenin pathway regulated genes for each treatment. Only the GSK3β inhibitor CHIR99021 significantly increased the transcript levels for each of these genes. (**H**–**J**) Scatter plots of pathway GO terms by two significance scores (pORA: *p*-value for over-representation analysis, pAcc: *p*-value from total perturbation accumulation for the pathway) corrected for false discovery rate. A data table for significant terms is listed in Appendix A. The yellow circle in each plot is the Wnt pathway. Red circle are significant GO-terms, and black circles are not significant. (**H**) 200 µM VPA, (**I**) 400 µM VPA, and (**J**) 10 µM CHIR99021. (**K**) Log2 fold change in transcript levels for 4 NTD-associated genes for each treatment. Both VPA and CHIR99021 decrease transcript levels for each of these genes.

**Figure 5 cells-12-01697-f005:**
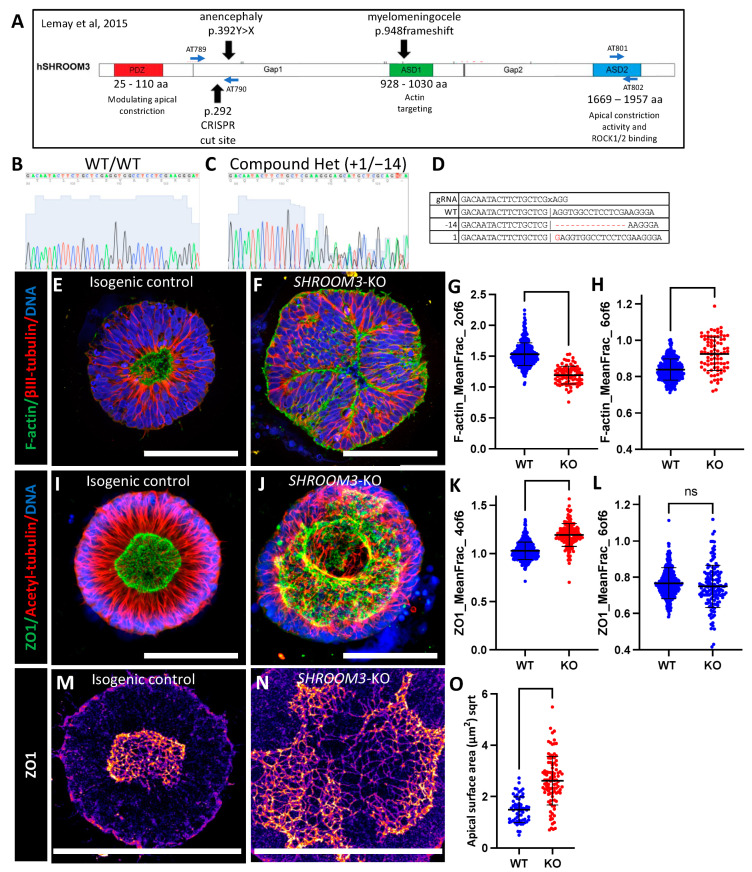
*SHROOM3* knockout SOSR-COs have enlarged dysmorphic lumens, basal f-actin, and impaired apical constriction. (**A**) Schematic of human *SHROOM3* gene with important locations, such as known human NTD variants, CRISPR cut site, and PCR primer locations [22]. (**B**,**C**) Sanger sequencing chromatographs for an isogenic control and compound heterozygous frameshift indel line using PCR primers AT789 and AT790. (**D**) The gRNA and indel sequences are listed. The “x” indicates the Cas9 cut site in the sequence. (**E**,**F**) Confocal images of SOSR-COs immunostained for β-III-tubulin and stained with f-actin dye and phalloidin-Alexa488. (**E**) Isogenic control has typical small round central lumen with radial tubulin structure. (**F**) *SHROOM3*-KO has enlarged dysmorphic lumen and basally localized f-actin. (**G**,**H**) Quantification of f-actin fluorescence radial distribution mean fraction in rings 2 and 6 compared with controls. n = 338 for WT and 83 for KO across 4 independent experiments. (**I**,**J**) Confocal images of SOSR-COs immunostained for acetyl-tubulin and ZO1. (**I**) Isogenic control has typical small round central lumen with radial tubulin structure. (**J**) *SHROOM3*-KO has enlarged dysmorphic lumen with large opening (max project is of the entire lumen Z-stack). (**K**,**L**) Quantification of ZO1 immunofluorescence radial distribution mean fraction in rings 4 and 6 compared with controls. n = 406 for WT and 164 for KO across 4 independent experiments. (**I**,**J**) Confocal images of SOSR-COs immunostained for acetyl-tubulin and ZO1. (**M**,**N**) Digital magnification of maximum Z-projection of one side of the lumen immunolabelled for ZO1 for isogenic control (**M**) and *SHROOM3*-KO (**N**). Fire LUT used to better see tight-junction edges. (**O**) Knockout cells have greater individual cell apical surface areas than isogenic controls. n = 61 for WT and 97 for KO. Each dot in (**G**,**H**,**K**,**L**) represents measurements for an individual SOSR-CO, while in (**N**), they represent individual apical cell surface areas. Bars are mean and standard deviation. Statistical analysis performed with Mann–Whitney test. Scale bars = 100 µm. ns *p* > 0.05.

**Figure 6 cells-12-01697-f006:**
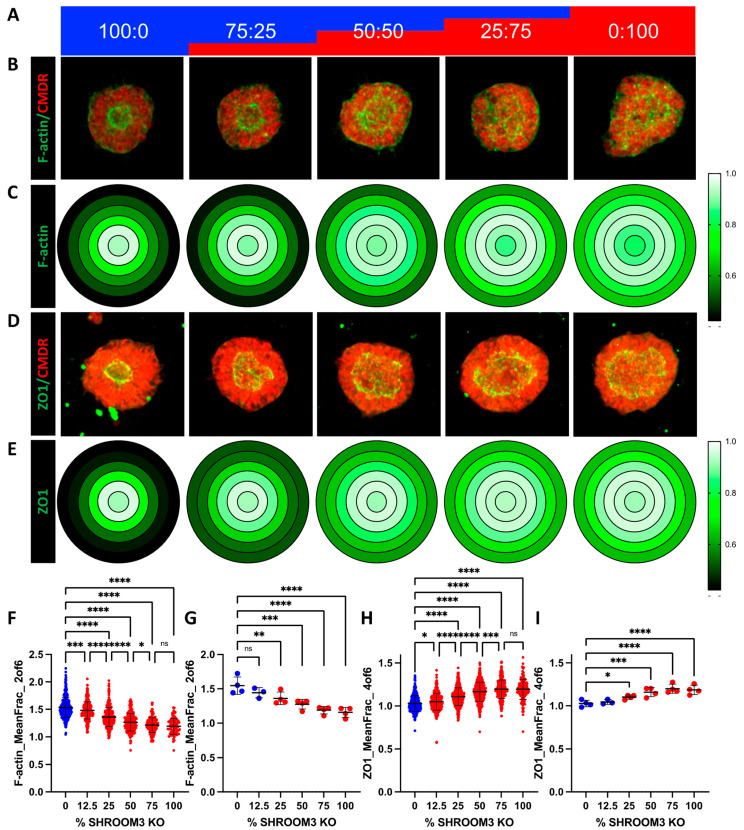
Mosaic mixing of WT and KO *SHROOM3* lines results in gene-dose response and suggests non-cell autonomous effects. (**A**) A schematic of the mixing of *SHROOM3*-KO and isogenic WT cells by percent. (**B**) Automated confocal maximum projections for exemplary SOSR-COs at each of the 5 mixes labelled for f-actin with phalloidin-Alexa488 and HCS CellMask Deep Red. (**C**) Average maximum normalized radial distribution of mean fraction of f-actin in each of the 6 concentric ring bins. Intensity is shown by 3 color heatmap, with white as the highest, black as the lowest, and green as intermediate. (**D**) Automated confocal maximum projections for exemplary SOSR-COs at each of the 5 mixes immunolabelled for ZO1 and HCS CellMask Deep Red. (**E**) Average maximum normalized radial distribution of mean fraction of ZO1 in each of the 6 concentric ring bins. Intensity is shown by 3 color heatmap, with white as the highest intensity, black as the lowest, and green as intermediate. (**F**–**I**) Comparing *SHROOM3*-KO distinguishing features across 6 different percentage doses of *SHROOM3* KO cells (0, 12.5, 25, 50, 75, and 100%; remainder of cells are WT). (**F**) Data for f-actin mean fraction of fluorescence radial distribution in ring 2. (**B**) Lumen solidity for each group. (**C**) The mean intensity of f-actin on the SOSR-COs basal edge was measured and a similar gene dose response was seen. n = 55, 68, 58, 59, and 22, respectively, across 2 independent experiments. Comparisons were not made with the Figure 5 dataset due to differences in overall fluorescence signal because of different phallodin-Alexa488 vendors. Bars are mean and standard deviation. Statistical analysis performed with Kruskal–Wallis with Dunn’s multiple comparisons test. * *p* < 0.05, ** *p* < 0.01, *** *p* < 0.001, **** *p* < 0.0001, ns *p* > 0.05.

## Data Availability

We deposited our RNA sequencing data files onto GEO (accession number GSE232218), and all other datasets are being deposited into FigShare. Jupyter notebooks containing custom image analysis pipelines in Python script can be found at: https://github.com/huangjoy1/sosr-analysis-files.

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
