# Peer review of "A Shared Pathogenic Mechanism for Valproic Acid and SHROOM3 Knockout in a Brain Organoid Model of Neural Tube Defects"

_cells, 2023, doi:10.3390/cells12131697_

Round 1

Reviewer 1 Report

Takla et al. has aimed to demonstrate clear, in vitro, dose-responsive structural phenotypes for acute valproic acid (VPA) treatment, characterize the transcriptomic effects of VPA, and identify inhibition of apical constriction via histone deacetylases inhibition as the likely mechanism by which VPA causes neural tube defects (NTDs). Moreover, they demonstrated altered structure, polarization, and reduced apical constriction for SHROOM3 knockout SOSRS, and they found that these phenotypes were gene-dose responsive in mosaic mixing experiments. The study is orginal, well prepared and discused. In my opinion this paper has a potential to make a positive contribution to the co-existing literature. I have only some minor recommendations for the authors:
1- I think that proprotein convertase subtilisin/kexin type 9 (PCSK9) may be involved in the etiopathogenesis of open NTDs at the critical steps of fetal neuronal differentiation. Although it has limitations, PCSK9 may be used as an additional biomarker for the screening of NTDs. The study mentioned below may contribute to this article.

“Erol SA, Tanacan A, Firat Oguz E, et al. A comparison of the maternal levels of serum proprotein convertase subtilisin/kexin type 9 in pregnant women with the complication of fetal open neural tube defects. Congenit Anom (Kyoto). 2021;61(5):169-176. doi:10.1111/cga.12432”

Author Response

We thank the reviewer for their positive reception of our manuscript. We were intrigued by the PCSK9 study and its potential utility as a biomarker for NTDs. However, in our model system there is low mRNA expression of PCSK9 (base mean is 149), and we see no significant change in expression at either 200 µM VPA (LFC:-0.286, padj: 0.177) or 400 µM VPA (LFC: -0.121, padj: 0.607). Interestingly, this gene does have increased expression in the GSK3b treated organoids (LFC: 1.087, pdaj:0.0221). One note is that this gene was seen as a biomarker in maternal serum, and, thus, its increased protein expression may not occur in the fetal tissue but in the maternal or extraembryonic (placental) tissues. We have now cited this publication as an example of biomarkers in NTDs which RNA-sequencing from multiple NTD etiologies in our model system may one day yield.

Reviewer 2 Report

This study uses SOSRS as a model for  neural tube closure defects. In this model, as known in other systems, a  reduction in apical constriction leading to a larger lumen can lead to the failure of the neural tube to close successfully. 

As a test case they use valproic acid, a small molecule that has previously been associated with NTD to show that it perturbs lumen size and apical constriction in their SOSRS model. As an example of gene defects associated with NTD they attempt to show that SHROOM3 also disrupts these same developmental modules. Thus they give a proof of concept for the use of SOSRS. 

Their observation that VPA and GSK3b could operate in parallel and cause a downregulation of SHROOM3, in turn a common mechanism for the disruption of apical constriction  is an interesting one, since they show that VPA in their model  is not acting through GSK3b. 

Major comments

1) I didn't see any evidence that they actually downregulated SHROOM3. Their supplementary figure (figure3 C and D)  could simply indicate that whatever off target effect,  it selectively removes the apical SHROOM3 staining while leaving the basal expression.  Other experiments are also consistent with an off target effect of their SHROOM3 gene disruption method. 

2) They state "Interestingly, 25% KO was significantly different from 0% for each measurement, but 75% and 100% were indistinguishable from one another for each measurement. This may suggest  a non-cell autonomous effect of SHROOM3 knockout as suggested in Xenopus studies[43]. " I didn't quite get why if the gene dosage effect plateaus at 75% that neccessarily indicates a non-cell autonomous effect. 

Author Response

Major comment responses

  1. We agree with the reviewers that the lack of solid evidence for the downregulation of SHROOM3 with indel formation is a weakness of our study. Standard qRT-PCR failed to show expression differences in transcript, and western blot analysis for 2 different antibodies lacked clearly identifiable SHROOM3 bands due to a large number of non-specific bands. It is known that the premature stop codons resulting from frameshift indels do not always result in nonsense mediated decay. This may be why we see the same amount of transcript in our KO line compared with the wild-type. For western blot, we have tried the 2 most referenced commercially available lines and were unable to identify a SHROOM3 band with either. By far, the most commonly used SHROOM3 antibody was a privately held antibody from the group that first found SHROOM3 to be an NTD gene in 2005. However, I have been unable to communicate with this group or identify another source for this antibody. A second common SHROOM3 antibody was discontinued by Santa Cruz Biotechnology.The immunostaining did show the loss of apical SHROOM3 immunostaining despite non-specific nuclear staining. Off-target effects of the SHROOM3 gRNA resulting in phenotypes already known to occur in SHROOM3-KO animal and cell models seems etremely unlikely. But in lieu of a better solution, we have put a sentence in our manuscript stating the limitation of our lack of KO confirmation and the unlikely but possible scenario that the phenotypes are due to an off-target effect.
  2. We were intrigued that many phenotypes were identifiable at 25% KO (and our new data presented in this revision shows significance at 12.5%) compared with 0% but no differences were seen between 75% and 100% KO. This could mean a non-cell autonomous effect of the KO cells on the WT cells. However, we agree with the reviewer that this is an overinterpretation of the current data and have removed this line from our revised manuscript. Future experiments will need to be performed to investigate this possibility directly.

Reviewer 3 Report

This manuscript reports an automated platform for the imaging and analysis of self-organizing single rosette spheroids.  This advance results in a robust in vitro human model to test genetic, environmental, and teratogenic causes of NTDs. This extremely rigorous and important study will have an important impact on the field.  I have only minor suggestions for improvement. 

The culture system makes it difficult to test for folic acid deficiency. Would it be possible to alter media conditions for folic acid deficiency? This may not be possible. However, with supplementation in the human population, most NTDs nowadays are not due to folate deficiency.  Along these lines, would treatment with Aminopterin + VPA shift the dose-response curve?

For the VPA experiments, the analysis focused on the area ratio, which had the greatest change in the random forest model.  There were 2 additional significant distinguishing features from random forest analysis. It was unclear why these features were not included in the univariate comparisons.   

There were a few instances where causal inferences were inappropriately made.  For instance, the statement that HDAC and GSK3b inhibition “mimics” VPA-induced changes in apical constriction is overstated.  Epistasis of similar analysis would be necessary to determine causality here.  Maybe better to say, “VPA, HDAC, and GSK3b inhibitors all results in a reduction in apical constriction” in the Figure 3 legend.  Also, on page 11 line 428, “…HDAC and GSK3b inhibition decrease apical constriction and one or both ate the likely potential mechanisms by which VPA causes NTDs.” Pg. 12 line 456 “These data show…”. Maybe replace “show” with “suggest” or “imply”

Interestingly, Shroom-3 knockout rosettes show incomplete closure. How frequently was this phenotype found?  Was this one of the parameters in Figure 5H-N?

Author Response

  1. We have not performed studies combining aminopterin and VPA. However, since high concentration aminopterin alone and folic acid supplementation with VPA had no effect on the organoid structure, we do not expect in the current model system that there would be an interaction with VPA. Our culture media has metabolites (hypoxanthine and thymidine) that can used via the salvage pathway to generate purines/nucleotides. Lack of folic acid and inhibitors of the folic acid pathway are thought to cause NTDs through lack of purines/nucleotides. Therefore, we plan in the future to obtain custom media without folic acid, hypoxanthine, and thymidine before testing the effects of aminopterin and folic acid again.Additionally, human studies have found that while folic acid supplementation reduces the risk of NTDs in the general population there is no evidence for reduced risk of NTDs with folic acid supplementation in VPA-exposed pregnancies (Jentink et al, 2010: https://doi.org/10.1002/pds.1975).
  2. The additional two distinguishing features for VPA concentration were both the ZO1-EGFP signal in the 4th concentric ring (mean fraction and fraction at distance). This data is depicted graphically in figure 2C,D along with the fluorescent intensity of the other 5 rings. However, we did not include statistical analysis in favor of this more intuitive heat-mapped intensity plot. We have added this quantitative data to this figure for the mean-fraction of EGFP signal in the 2-4th rings in Figure 2e-g.
  3. We appreciate the reviewer’s suggestion about word choice when comparing VPA, HDAC, and GSK3b and have changed the wording to fit their suggestions.
  4. In our original analysis pipeline that utilized non-confocal epifluorescence (Figure 3 and S2C), it was impossible to determine if the lumen had an opening. Likewise, our current high-throughput confocal platform generates maximum projections from all optical sections to ensure the lumen and SOSRS are depicted at their largest. Identifying these openings with our platform would likely necessitate full 3D reconstructions. With our automated imaging pipeline, this would necessitate keeping TB of original source data while manual confocal would take possible 10-100’s of hours to reach a statistical conclusion. We are working with co-author Dr. Jonathan Sexton to beta-test a possible analysis pipeline, but unfortunately, we did not keep most of our raw image data (before maximum projection) to conserve data storage space. We agree with the reviewer that although more challenging to identify, this phenotype is directly related to the disease pathology (open neural tube), and we must attempt to address in our future studies.

Additional changes in this revised manuscript:

  • We have changed the name from SOSRS to SOSR-COs (self-organizing single rosette cortical organoids) to fit nomenclature rules suggested by leaders in the brain organoid field: https://doi.org/10.1038/s41586-022-05219-6).
  • Figure 5 has additional SHROOM3 apical surface measurement data and better images to show apical surface area comparison between WT and KO. Figure 5 also has WT vs. KO data from 4 plate replicates (rather than only 2 as in our previous manuscript) (Figure 5G,H,K,L). We also changed from the f-actin signal on the edge of the SOSR-COs to the fluorescence in the 6th ring (Figure 5H). This was significantly increased with f-actin but was unchanged for ZO1 (Figure 5L) showing increased basal localization of f-actin in SHROOM3-KO SOSR-COs.
  • Combined 4 plates for SHROOM3 random forest model data (original was from 2 plates) and graphed the most distinguishing feature for f-actin and ZO1 each. (Figure S4).
  • Figure 6 now has 4 plates of data and plots the most distinguishing feature for f-actin (ring 2 radial distribution, Fig 6F) and ZO1 (ring 4 radial distribution, Fig 6H). We also plot the mean values for each of the plates (Fig 6G,I).
  • We have deposited the RNA-sequencing data in GEO and provide the link and reviewer token here. To review GEO accession GSE232218:

https://nam02.safelinks.protection.outlook.com/?url=https%3A%2F%2Fwww.ncbi.nlm.nih.gov%2Fgeo%2Fquery%2Facc.cgi%3Facc%3DGSE232218&data=05%7C01%7Cdamki%40med.umich.edu%7C5ab859fb9917456ce60f08db519fd7cf%7C1f41d613d3a14ead918d2a25b10de330%7C0%7C0%7C638193519182630866%7CUnknown%7CTWFpbGZsb3d8eyJWIjoiMC4wLjAwMDAiLCJQIjoiV2luMzIiLCJBTiI6Ik1haWwiLCJXVCI6Mn0%3D%7C3000%7C%7C%7C&sdata=sodE2aj56ofv6XpBY4NKX3qukxyUX%2FKgzJ55hJu86wU%3D&reserved=0

Enter token mjefmceuxduthkv into the box